# Diverse modulatory effects of bibenzyls from *Dendrobium* species on human immune cell responses under inflammatory conditions

Virunh Kongkatitham[1,2☯], Adeline Dehlinger[3☯], Chatchai Chaotham[4], Kittisak Likhitwitayawuid[2], Chotima Böttcher [3‡*], Boonchoo Sritularak[2,5‡*]

**1** Pharmaceutical Sciences and Technology Program, Faculty of Pharmaceutical Sciences, Chulalongkorn University, Bangkok, Thailand, **2** Department of Pharmacognosy and Pharmaceutical Botany, Faculty of Pharmaceutical Sciences, Chulalongkorn University, Bangkok, Thailand, **3** Experimental and Clinical Research Center, a Cooperation Between the Max Delbrück Center for Molecular Medicine in the Helmholtz Association and Charité – Universitätsmedizin Berlin, Berlin, Germany, **4** Department of Biochemistry and Microbiology, Faculty of Pharmaceutical Sciences, Chulalongkorn University, Bangkok, Thailand, **5** Center of Excellence in Natural Products for Ageing and Chronic Diseases, Faculty of Pharmaceutical Sciences, Chulalongkorn University, Bangkok, Thailand

☯ These authors contributed equally to this work.
‡ CB and BS jointly directed this work.
* chotima.boettcher@charite.de (CB); boonchoo.sr@chula.ac.th (BS)

**Data Availability Statement:** All relevant data are within the paper and its Supporting information files.

## Abstract

*Dendrobium* plants are widely used in traditional Chinese medicine. Their secondary metabolites such as bibenzyls and phenanthrenes show various pharmacological benefits such as immunomodulation and inhibitory effects on cancer cell growth. However, our previous study also showed that some of these promising compounds (i.e., gigantol and cypripedin) also induced the expression of inflammatory cytokines including TNF in human monocytes, and thus raising concerns about the use of these compounds in clinical application. Furthermore, the effects of these compounds on other immune cell populations, apart from monocytes, remain to be investigated. In this study, we evaluated immunomodulatory effects of seven known bibenzyl compounds purified from *Dendrobium* species in human peripheral blood mononuclear cells (PBMCs) that were stimulated with lipopolysaccharide (LPS). Firstly, using flow cytometry, moscatilin (**3**) and crepidatin (**4**) showed the most promising dose-dependent immunomodulatory effects among all seven bibenzyls, determined by significant reduction of TNF expression in LPS-stimulated CD14$^+$ monocytes. Only crepidatin at the concentration of 20 μM showed a significant cytotoxicity, i.e., an increased cell death in late apoptotic state. In addition, deep immune profiling using high-dimensional single-cell mass cytometry (CyTOF) revealed broad effects of *Dendrobium* compounds on diverse immune cell types. Our findings suggest that to precisely evaluate therapeutic as well as adverse effects of active natural compounds, a multi-parameter immune profiling targeting diverse immune cell population is required.

**Funding:** This research project is supported by Second Century Fund (C2F), Chulalongkorn University to V.K. and B.S. and funded by Thailand Science research and Innovation Fund Chulalongkorn University (CU_FRB65_hea (57) _066_33_10). V.K. is grateful to C2F for conducting research abroad scholarship, Chulalongkorn University for a Ph.D. research abroad. The funders had no role in study design, data collection and analysis, decision to publish, or preparation of the manuscript.

**Competing interests:** The authors have declared that no competing interests exist.

## Introduction

Orchidaceae is one of the most prominent families of flowering plants with approximately 25,000 species known worldwide [1]. *Dendrobium* is one of the largest genera in the orchid family with more than 1,500 species. *Dendrobium* can be found in a wide area including tropical and subtropical Asia and Oceania region [2, 3]. In China, some of the native *Dendrobium* species have been widely used in the traditional Chinese medicine, and significantly contribute to the growth of the industrial use of medicinal plants [4]. In addition, *Dendrobium (*also known as Shihu*)* has been also used as an important food ingredient in many dietary supplements such as Shihu wine and Fengdou Shihu [5]. Secondary metabolites from *Dendrobium* such as flavonoids, bibenzyls, phenanthrenes, alkaloids and sesquiterpenoids have been reported to provide various pharmacological activities, for instance, anti-inflammatory, antioxidant, antiangiogenic, anticancer, antimicrobial, neuroprotective and immunomodulatory activities [6–20]. However, most of these studies were performed in human cell lines or animal models [15–20]. Only a small number of studies utilized primary human cell cultures, for example, our previous study on immunomodulatory effects of a bibenzyl compound (4,5-dihydroxy-3,3´,4´-trimethoxybibenzyl) isolated from *D. lindleyi* Steud. in CD14$^+$ monocytes under inflammatory conditions [9]. Our findings have also demonstrated that treatment with *Dendrobium* bibenzyls gigantol and cypripedin could also induce the expression of inflammatory cytokines, including tumor necrosis factor (TNF) and interleukin 6 (IL-6), in CD14$^+$ monocytes. These results suggested potential adverse effects of some bibenzyl compounds on primary human immune cells. Therefore, it is important to evaluate the therapeutic potential, mechanism of action and potential adverse effects of natural active compounds in human system before considering their application in clinical settings.

Inflammation is a defense mechanism of the body against various stimuli such as pathogens, toxic substances or damaged cells [21]. During inflammation, innate immune cells, including dendritic cells (DCs), neutrophils, monocytes and macrophages, interact with exogenous or endogenous molecules to mediate inflammatory responses [22]. These cells express receptors such as toll-like receptors 9 (TLR9) which recognize DNA from damaged tissues, known as danger-associated molecular patterns (DAMPS), or TLR4 for pathogen-associated molecular patterns (PAMPs) such as lipopolysaccharide (LPS) [23–25]. LPS, an outer membrane substance of gram-negative bacteria, is widely used in *in vitro* functional assay to induce inflammatory conditions [26]. LPS binds to CD14, a glycosylphosphatidylinositol (GPI)-linked surface protein which is mostly expressed on myeloid cells and transferred to TLR4 complex, resulting in increased expression of inflammatory cytokines such as TNF, interleukin-2 (IL-2) and interferon-gamma (IFN-γ) [27–29]. Such responses occur for example via intracellular phosphor-molecules such as phosphorylated extracellular signal-regulated kinase 1/2 (pERK1/2), phosphorylated signal transducer and activator of transcription 1 and 5 (pSTAT1 and pSTAT5) [30].

In this study, we evaluated immunomodulatory effects of seven known bibenzyls derived from *Dendrobium* plants on multiple human immune cells using flow cytometry and high-dimensional mass cytometry (CyTOF). Our results demonstrated that all seven bibenzyls could reduce the expression of TNF in CD14$^+$ monocytes, induced by LPS treatment. However, moscatilin and crepidatin showed the most promising dose-dependent effects. Further deep immune profiling revealed that moscatilin and crepidatin also modulated responses of non-classical monocytes and nature killer cells, apart from CD14$^+$ monocytes. Our results underline the need for multi-parameter immune profiling methods (such as CyTOF) to precisely assess therapeutic potentials and adverse effects of bibenzyl compounds.

## Materials and methods

### Plant materials

The whole plants of *Dendrobium scabrilingue* Lindl., *Dendrobium capillipes* Rchb.f., *Dendrobium secundum* (Blume) Lindl. and *Dendrobium signatum* Rchb. f. were purchased from Jatuchak market, Bangkok [31–33]. *D. scabrilingue*, *D. secundum*, *D. capillipes* and *D. signatum* were authenticated by comparison with herbarium specimens at the Department of National Park, Wildlife and Plant Conservation, Ministry of Natural Resources and Environment [31–33]. The voucher specimens of *D. scabrilingue* (BS-DScab-12255), *D. secundum* (DS/BS-092552), *D. capillipes* (DC-082553) and *D. signatum* (BS-DS-102555) have been deposited at the Department of Pharmacognosy and Pharmaceutical Botany, Faculty of Pharmaceutical Sciences, Chulalongkorn University [25–27].

### Compounds and reagents

Seven bibenzyls (Fig 1) were isolated from *Dendrobium* plants, as previously described (Fig 2) [31–33]. The ethyl acetate (EtOAc) extract from *Dendrobium scabrilingue* Lindl. was subjected to vacuum-liquid chromatography (VLC) over silica gel using EtOAc-hexane, gradient to give 8 fractions (A-H). Fraction D was fractionated by column chromatography (CC) on silica gel (EtOAc-hexane, gradient) to obtain 14 fractions (DI-DXIV). Fraction DIX was purified by Sephadex LH-20 (MeOH) and then separated by CC (silica gel, EtOAc-CH$_2$Cl$_2$, gradient) to give batatasin III (**1**). Aloifol I (**7**) was obtained from fraction DX after purification on Sephadex LH-20 (MeOH) and CC (silica gel, CH$_2$Cl$_2$) [31]. The methanol (MeOH) extract from *Dendrobium secundum* was separated by VLC over silica gel (EtOAc-hexane and MeOH-CH$_2$Cl$_2$, gradient) to give 8 fractions (A-H) Fraction G was fractionated by VLC on

**Fig 1. Chemical structures of seven known bibenzyls from *Dendrobium* species.**

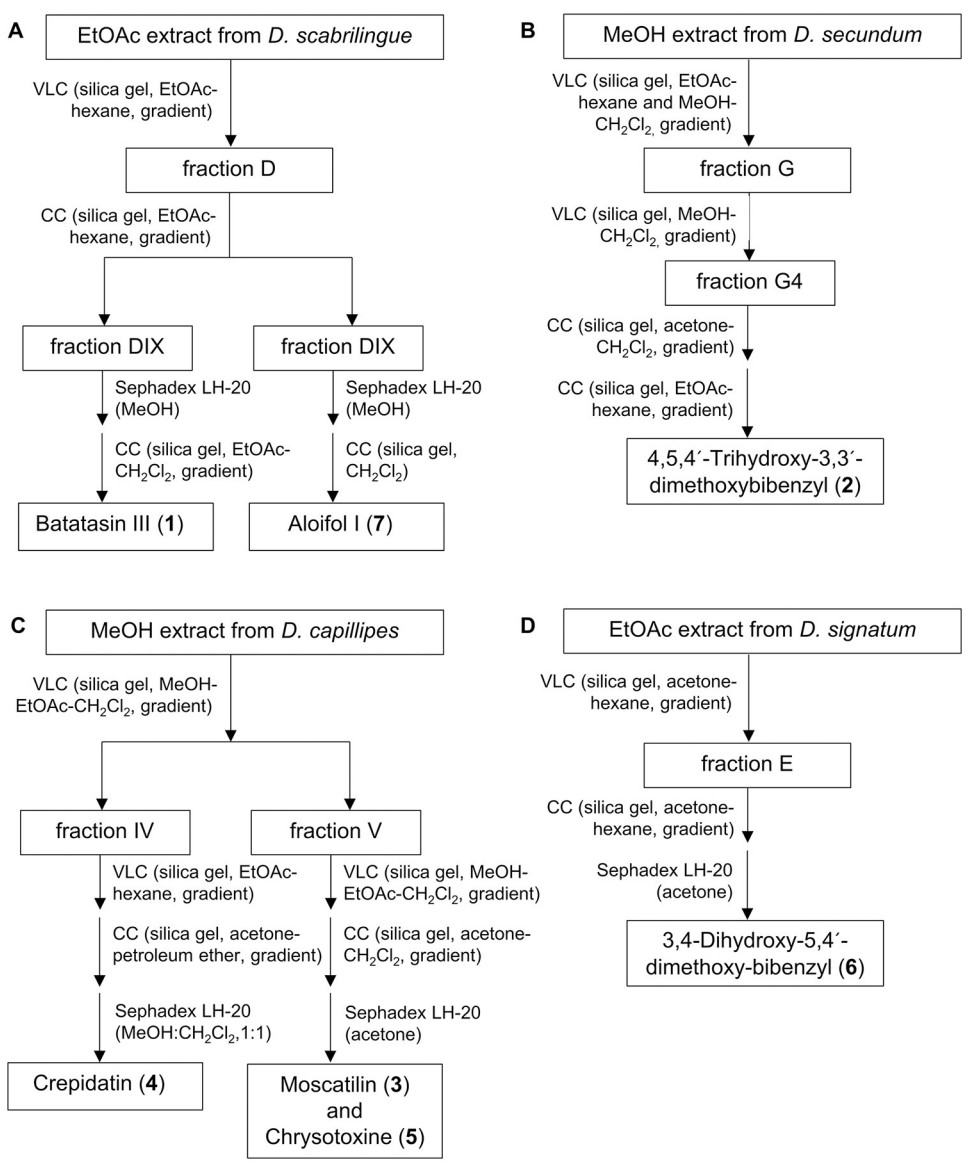

**Fig 2. The schematic diagram for compound purification from *Dendrobium* species.** (A) Batatasin III and aloifol I isolated from *D. scabrilingue*. (B) 4,5,4′-trihydroxy-3,3′-dimethoxybibenzyl isolated from *D. secundum*. (C) Moscatilin, crepidatin and chrysotoxine isolated from *D. capillipes*. (D) 3,4-Dihydroxy-5,4′-dimethoxy-bibenzyl isolated from *D. signatum*.

silica gel (MeOH-CH$_2$Cl$_2$, gradient) to obtain six fractions (G1-G6). Fraction G4 was further separated by CC (silica gel, acetone-CH$_2$Cl$_2$, gradient) and purified by CC (EtOAc-hexane, gradient) to yield 4,5,4′-trihydroxy-3,3′-dimethoxybibenzyl (**2**) [32]. The MeOH extract from *Dendrobium capillipes* was subjected to VLC on silica gel (MeOH-EtOAc-CH$_2$Cl$_2$, gradient) to give 7 fractions (I-VII). Fraction IV was fractionated by VLC over silica gel (EtOAc-hexane, gradient) to obtain 13 fractions (IV-A to IV-J). Fraction IV-J was separated by CC (silica gel, acetone-petroleum ether, gradient) and further purified on Sephadex LH-20 (MeOH-CH$_2$Cl$_2$, 1:1) to yield crepidatin (**4**). Moscatilin (**3**) and chrysotoxine (**5**) were obtained from fraction V after separation by VLC over silica gel using gradient elution of MeOH-EtOAc-CH$_2$Cl$_2$-

hexane, further separation by CC (silica gel, acetone-$CH_2Cl_2$, gradient) and purification on Sephadex LH-20 (acetone) [32]. The EtOAc extract from *Dendrobium signatum* Rchb. f. was subjected to VLC on silica gel (acetone- hexane, gradient) to give 8 fractions (A-H). 3,4-Dihydroxy-5,4′-dimethoxy-bibenzyl (**6**) was yielded from fraction E after fractionation by CC (silica gel, acetone-hexane, gradient) and purification on Sephadex LH-20 (acetone) [33]. Dimethyl sulfoxide (DMSO), LPS and brefeldin A were purchased from Sigma Aldrich (St. Louis, MO, USA). Roswell Park Memorial Institute (RPMI) 1640, fetal bovine serum (FBS), phosphate buffered saline (PBS), SMART TUBE INC Proteomic Stabilizer and 16% w/v formaldehyde (FA) were purchased from Thermo Fisher Scientific Inc. (Rockford, IL, USA). Anti-human antibodies were purchased pre-conjugated to metal isotopes (Fluidigm) or conjugated in house following the manufacturer's protocol by using the MaxPar X8 kit (Fluidigm).

## Ethics and cells

This study was approved by the Ethics Committee of Charité–Universitätsmedizin Berlin. Buffy coats from three healthy blood donors were obtained from the German Red Cross (GRC) for research use. Human PBMCs were isolated from three healthy donors (male, at the age of 29, 30 and 44 years old). PBMCs were subsequently aliquoted at $20 \times 10^6$ cells/mL and cryopreserved in liquid nitrogen tank, as described in the previous study [9].

## LPS stimulation in human PBMCs

Frozen PBMCs were thawed and resuspended in RPMI 1640 medium with 10% FBS and cell concentration was adjusted to $20 \times 10^6$ cells/mL. Cells were plated in an ultralow-attachment 96-well plate (Corning, New York, USA) at a density of $5 \times 10^5$ cells per well. Four different concentrations of compounds were then added to the corresponding well. For cell stimulation, a final concentration of 100 ng/mL of LPS was added into the cultured cells. After a 2 h incubation, a total concentration of 10 μg/mL of brefeldin A was added into the wells and further incubated for another 2 h. Cells were then harvested into 1.5 mL microtubes and washed with PBS. Finally, cells were resuspended in 10% BSA and incubated with SMART TUBE INC Proteomic Stabilizer for 12 min at RT. Stabilized cells were stored at -80˚C before staining.

## Flow cytometry

Cells were thawed, washed twice and transferred into 1.5 mL microtubes. Cells were incubated in FcR-blocking buffer (1:100; Miltenyi Biotec, Bergisch Gladbach, Germany) at 4˚C for 10 min to block unspecific antibodies binding to Fc receptors. Cells were incubated for 20 min at 4˚C with fluorochrome-conjugated extracellular antibodies for CD3 (APC, HIT3a, Biolegend), CD14 (FITC, RMO52, Beckman Coulter), CD19 (PE, HIB19, Biolegend) and HLA-DR (APC/Cy7, L243, Biolegend) diluted in staining buffer (0.5% BSA in PBS containing 2mM EDTA). Cells were washed with staining buffer and were then fixed with 2% methanol-free FA at 4˚C for 30 min. After washing with staining buffer, cells were incubated for 30 min at 4˚C with fluorochrome-conjugated antibodies for intracellular proteins including TNF (brilliant violet, MAb11, Biolegend), IL-2 (PerCP/Cy5.5, MQ1-17H12, Biolegend) and IFN-γ (PE/Cy7, 4S.B3, Biolegend) diluted in permeabilization buffer (eBioscience, California, USA). Furthermore, cells were washed with staining buffer and were fixed with 4% methanol-free FA at 4˚C for 10 min, then washed with staining buffer and centrifuged at 600 x g at 12˚C for 5 min. Subsequently, pellets were resuspended in staining buffer and were acquired on BD CANTO II flow cytometer (BD Biosciences, New Jersey, USA) with software BD DIVA version 8.1. Data analysis was performed using FlowJo software version 10.1 (Ashland, OR, USA).

## CyTOF measurement

For phosphoproteins measurement, cells were incubated with 100 ng/mL of LPS for 15 min. Cells were incubated with Cisplatin-195Pt (1:3000) for 1 min at RT and then fixed with 16% methanol-free FA. Cells were harvested into 1.5 mL microtubes and washed with PBS. Finally, cells were resuspended in 10% BSA and incubated with SMART TUBE INC Proteomic Stabilizer for 12 min at RT. Stabilized cells were stored at -80˚C before staining. Cells were stained and analysed according to our standard protocol [34].

**Intracellular Barcoding.** After fixation and storage at -80˚C, cells were thawed and subsequently stained with premade combinations of the palladium isotopes $^{102}$Pd, $^{104}$Pd, $^{105}$Pd, $^{106}$Pd, $^{108}$Pd and $^{110}$Pd (Cell-ID 20-plex Pd Barcoding Kit, Fluidigm). Each sample received a unique combination of three different palladium isotopes. Therefore, it was possible to generate up to twenty different barcodes. One sample did not receive a barcode allowing to increase the sample size to 21 samples. Cells were stained with the barcodes for 30 min at RT and then washed twice with cell staining buffer. The 21 samples were pooled together, washed and further stained with antibodies.

**Antibody staining.** Samples were pooled, then resuspended in 50 µL of antibody cocktail against surface markers and incubated for 30 min at 4˚C. Cells were washed twice with staining buffer and subsequently fixed overnight with 2% methanol-free FA solution. Fixed cells were washed with staining buffer, then permeabilized with 100 µL ice-cold methanol for 10 min at 4˚C. Cells were washed twice in staining buffer and resuspended with 100 µL of antibody cocktail against phosphor-protein markers. After 30 min of incubation at RT, samples were washed twice with staining buffer and resuspended in 1 mL of iridium mix (1:1000 Iridium in PBS containing 2% FA) for 30 min at RT. Cells were washed twice with staining buffer and kept at 4˚C until CyTOF measurement. All antibodies used are listed in Table 1.

**Mass cytometry data processing and analysis.** Initial manual gating of CD45$^+$DNA$^+$ and gating out of CD3$^+$CD19$^+$ cells and de-barcoding according to the barcode combination were performed on FlowJo. De-barcoded samples were exported as individual FCS files for further analysis. Using the R package CATALYST, each file was compensated for signal spillover. Using FlowJo, dead cells, which are Cisplatin$^+$, were gated out and FCS files were exported. For further analysis, we used previously described scripts and workflows. We created multi-dimensional scaling (MDS) plots on median marker expression from all markers for first evaluation of the overall similarities between samples and conditions. In order to perform unsupervised clustering, we used the FlowSOM/ConsensusClustserPlus algorithms of the CATALYST package. We opted for a total number of 20 meta clusters based on the phenotypic heatmaps and the delta area plot. We generated UMAP representations including all markers as input in order to have a dimensionality reduction visualization of the clusters.

## Cytotoxicity determined by Annexin V and 7-AAD staining in human PBMCs

Frozen PBMCs were resuspended in RPMI 1640 medium with 10% FBS and cell concentration was adjusted to 20 x 10$^6$ cells/mL. Cells were seeded at a density of 5 x 10$^5$ cells/10 µL in each well of an ultra-low attachment 96-well plate. Different concentrations of compounds were then added to the corresponding wells and incubated for 4 h. Cells were harvested, washed and transferred into 1.5 mL microtubes. Cells were subsequently incubated in CD45 antibody (APC, HI30, Biolegend) diluted in staining buffer at 4˚C for 20 min. After washing, cells were resuspended with 100 µL of Annexin V binding buffer. Cell suspensions were then transferred into new microtubes, and further incubated with Pacific Blue™ Annexin V Apoptosis Detection Kit with 7-AAD (Biolegend) at RT for 15 min in the dark. Annexin V binding buffer was

**Table 1. The CyTOF antibody list.**

| Target | Isotope tag | Clone | Company |
|---|---|---|---|
| CD45 | $^{86}$Y | HI30 | Fluidigm |
| HLADR | $^{141}$Pr | L243 | BioLegend |
| CD19 | $^{142}$Nd | HIB19 | Fluidigm |
| p53 | $^{143}$Nd | 7F5 | Fluidigm |
| CD69 | $^{144}$Nd | FN50 | Fluidigm |
| CD4 | $^{145}$Nd | RPA-T4 | Fluidigm |
| CD64 | $^{146}$Nd | 10.1 | Fluidigm |
| pH2AX | $^{147}$Sm | JBW301 | Fluidigm |
| CD16 | $^{148}$Nd | 3G8 | Fluidigm |
| CD56 | $^{149}$Sm | NCAM16.2 | BD Biosciences |
| pSTAT5 | $^{150}$Nd | 47 | Fluidigm |
| ICOS | $^{151}$Eu | C398.4A | Fluidigm |
| pAKT | $^{152}$Sm | D9E | DVS Science |
| pSTAT1 | $^{153}$Eu | 58D6 | Fluidigm |
| CD3 | $^{154}$Sm | UCHT1 | Fluidigm |
| CD11c | $^{155}$Gd | Bu15 | BioLegend |
| CD86 | $^{156}$Gd | IT2.2 | Fluidigm |
| pSTAT3 | $^{158}$Gd | 4/P-Stat3 | Fluidigm |
| CD1c | $^{159}$Tb | L161 | BioLegend |
| CD14 | $^{160}$Gd | RM052 | Fluidigm |
| CTLA4 | $^{161}$Dy | 14D3 | Fluidigm |
| CD8 | $^{162}$Dy | RPA-T8 | Fluidigm |
| CRTH2 | $^{163}$Dy | BM16 | Fluidigm |
| Ikba | $^{164}$Dy | L35A5 | Fluidigm |
| pCREB | $^{165}$Ho | 87G3 | DVS Science |
| pnFkBp65 | $^{166}$Er | K10895.12.50 | Fluidigm |
| CCR7 | $^{167}$Er | G043H7 | Fluidigm |
| CCR9 | $^{168}$Er | L053E8 | Fluidigm |
| CD33 | $^{169}$Tm | WM53 | Fluidigm |
| Tbet | $^{170}$Er | 4B10 | BioLegend |
| pERK1/2 | $^{171}$Yb | D13.14.4E | Fluidigm |
| CX3CR1 | $^{172}$Yb | 2A9-1 | BioLegend |
| CXCR4 | $^{173}$Yb | 12G5 | Fluidigm |
| PD1 | $^{174}$Yb | EH12.2H7 | Fluidigm |
| pS6 | $^{175}$Lu | N7-548 | Fluidigm |
| CD11b | $^{209}$Bi | ICRF44 | Fluidigm |

added to each cell suspension. Stained cells were acquired on BD CANTO II flow cytometer (BD Biosciences, New Jersey, USA) with software BD DIVA version 8.1. The obtained data were analysed using FlowJo software version 10.1 (Ashland, OR, USA).

## Statistical analysis

Data, expressed as mean ± standard deviation (SD), were analysed for statistical significance ($p < 0.05$) using one-way ANOVA with Tukey's test in GraphPad Prism v.9.0 software (San Diego, CA, USA), unless otherwise stated.

## Results

### Immune modulatory effects of bibenzyls from *Dendrobium* species on primary human immune cells

First, to induce inflammatory conditions *in vitro*, we incubated human PBMCs in the presence of LPS, as previously described [9]. After 4 h of LPS stimulation, we detected an increased expression of TNF in CD14+ monocyte population, whereas IL-2 and IFN-γ expression were unchanged in this population. Furthermore, TNF expression was not induced in other immune cell types, showing specific responses of monocytes to LPS (Fig 3).

Second, to evaluate dose-dependent immunomodulatory effects, LPS-stimulated PBMCs were treated with seven known bibenzyl compounds (Fig 1) isolated from *Dendrobium* plants (Fig 2) including batatasin III (**1**), 4,5,4´-trihydroxy-3,3´-dimethoxybibenzyl (**2**), moscatilin (**3**), crepidatin (**4**), chrysotoxine (**5**), 3,4-dihydroxy-5,4´-dimethoxy-bibenzyl (**6**) and aloifol I (**7**) at four different concentrations. These four known concentrations (1, 5, 10 and 20 μM) have been previously used to evaluate therapeutic potentials of the compounds [9, 12]. We also controlled DMSO effects by treating LPS-stimulated PBMCs with four different concentrations of DMSO, which were the same concentrations used for diluting the compounds. Untreated PBMCs (i.e., no LPS and no compound treatment) were used as a control. We detected a significant reduction in the frequency of TNF-expressing CD14+ monocytes treated with all bibenzyl compounds, except batatasin III (**1**) (Fig 4). This effect was not found in other immune cell populations, as well as in DMSO-treated condition. No changes in LPS-induced IL-2 and IFN-γ expression were detected in monocytes and other immune cell types, as well as in DMSO-treated PBMCs. Nevertheless, only three compounds (4,5,4´-trihydroxy-3,3´-dimethoxybibenzyl (**2**), moscatilin (**3**) and crepidatin (**4**)) exhibited inhibitory effects in a dose-dependent manner (Fig 4). However, we observed a sign of instability of compound **2** under our experimental conditions, e.g., changes in the color of culture medium, possibly due to multiple hydroxyl groups, especially the ortho-hydroxy. Of note, the compound **2** showed also a slight increase of TNF-expressing monocytes at the concentration of 20 μM. To prove the stability of the compound **2**, further evaluation will be required. Therefore, in this study, we selected compounds **3** and **4** for further investigations.

### Investigation of potential cytotoxicity of moscatilin and crepidatin

We next proved whether the reduction of TNF-expressing monocytes mentioned above was not resulted by decreased cell numbers due to the cytotoxicity of the compound **3** and **4**. To do so, we determined the apoptotic states of human PMBCs after 4 h incubation in the presence of the compound **3** and **4** at four different concentrations. Similar to the above assessment, DMSO treatment was also tested for cytotoxicity. No significant differences in the frequency of live cells (CD45+Annexin V-7-AAD-) or the frequency of cells with early (CD45+Annexin V+7-AAD-) or late apoptotic (CD45+Annexin V+7-AAD+) state could be detected, after treatment with moscatilin (**3**) and DMSO (Fig 5). However, we detected significant increase in the frequency of cells at late apoptotic state after treatment with 20 μM of crepidatin (**4**) (Fig 5). Considering these findings, we decided to use the concentration of 10 μM for both moscatilin and crepidatin in further deep immune profiling using mass cytometry.

### Deep immune profiling revealed a broad spectrum of immunomodulatory effects of moscatilin and crepidatin

To further characterize the modulatory effects of moscatilin and crepidatin on a wide spectrum of immune cell types, as well as their immunomodulatory effects under LPS-induced

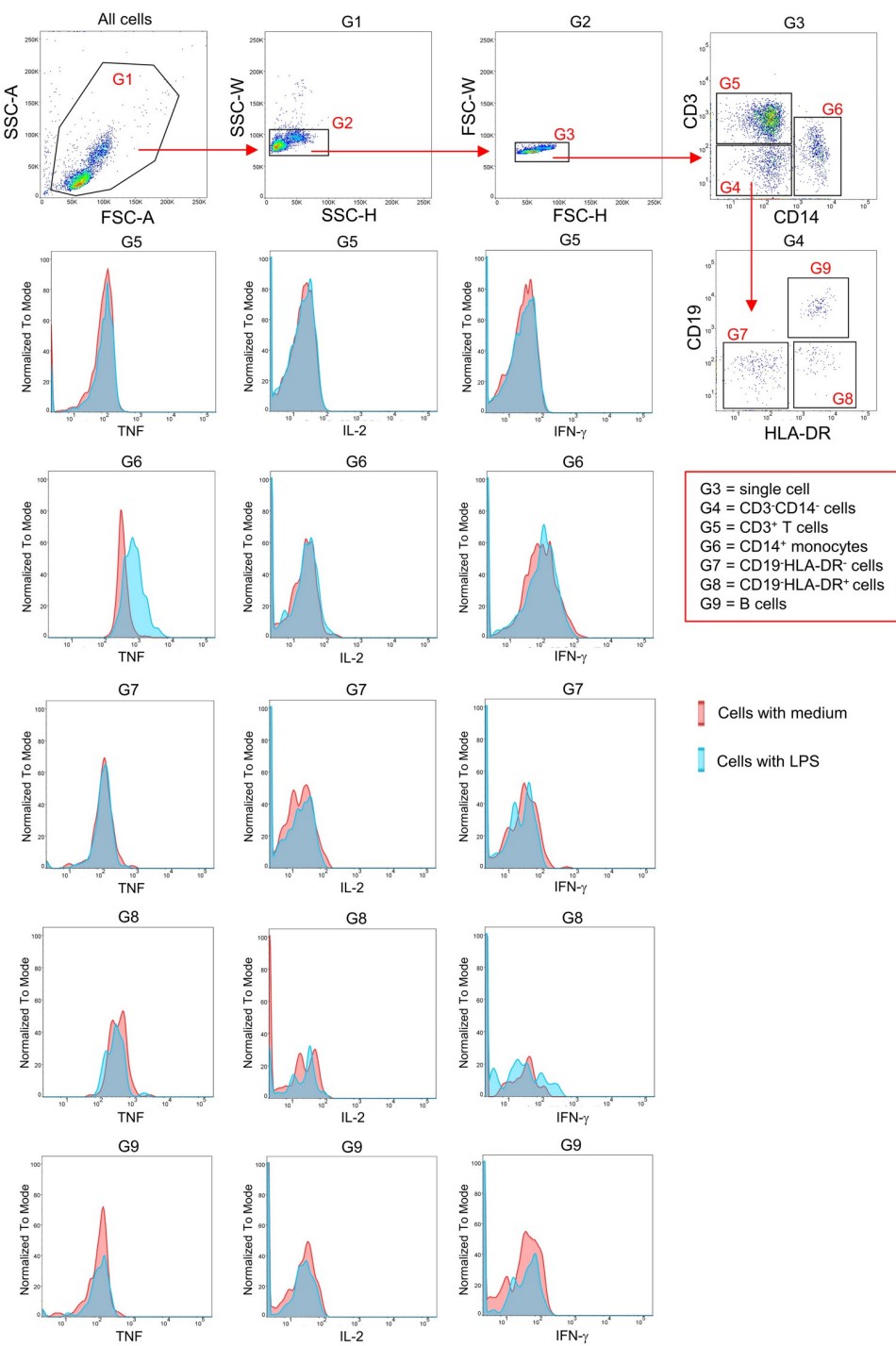

**Fig 3. Flow cytometry analysis.** Dot plots exhibit gating strategy of human CD3⁺ T cells (G5), CD14⁺ monocytes (G6), CD19⁻HLA-DR⁻ (G7), CD19⁻HLA-DR⁺ (G8) and B cells (G9).

inflammation, we applied our previously validated immune profiling workflow using CyTOF [28]. PBMCs from three healthy donors were incubated with either LPS, an active compound (moscatilin or crepidatin) or a combination of LPS and an active compound. After 4 hours incubation, we stained the samples with an antibody panel targeting 37 protein markers

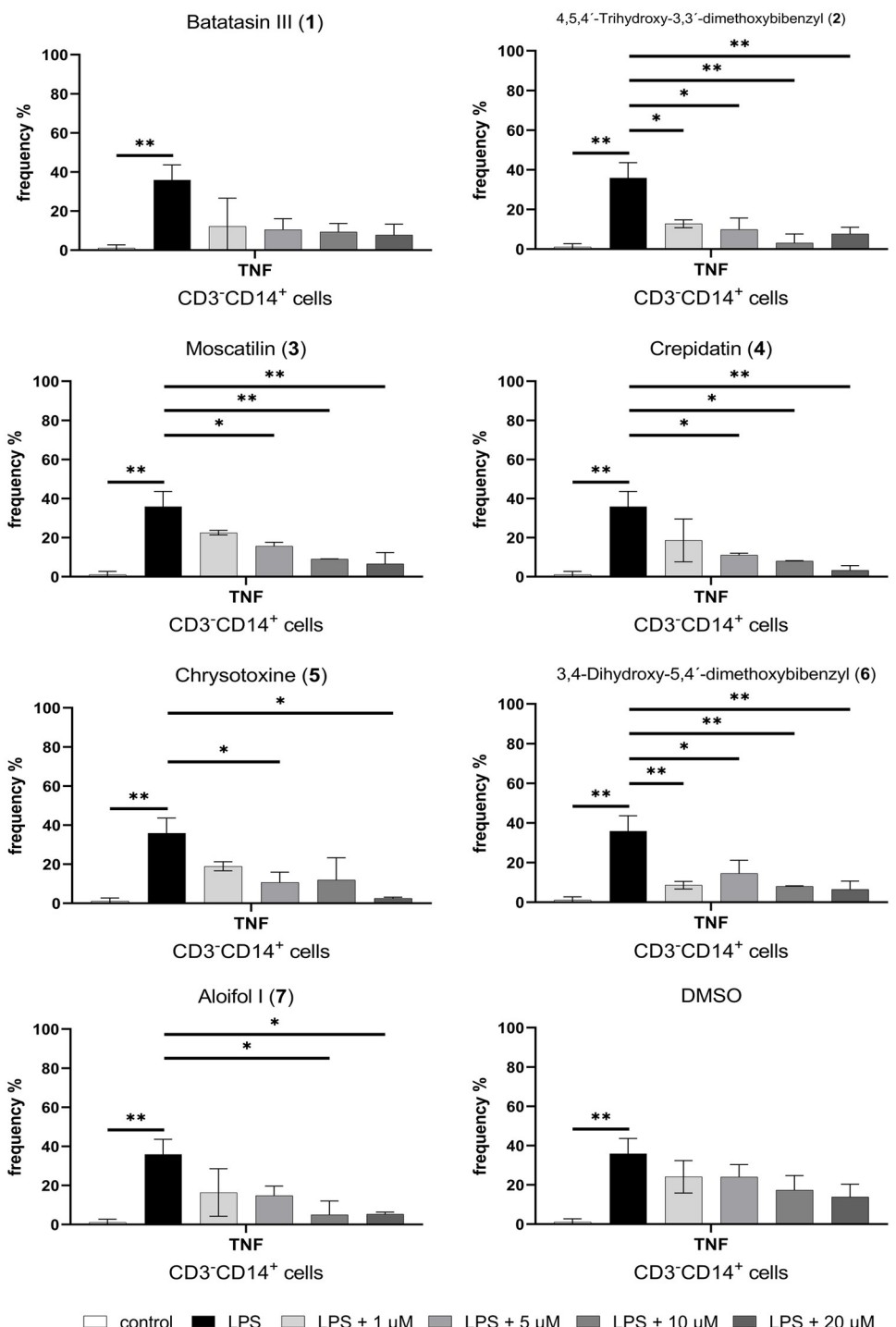

**Fig 4. Bar graphs show the mean frequency (% of total CD45$^+$ cells) of inflammatory cytokines (TNF-α, IL-2 and IFN-γ) expression in T cells, monocytes and B cells after 4 h incubation at different conditions, which are 1) Control = without LPS and compound; 2) LPS = with LPS stimulation but no compound treatment; or 3) with LPS stimulation and different concentration of seven known bibenzyl compounds, i.e., 1, 5, 10 or 20 μM.** One-way ANOVA, corrected for multiple comparisons by Tukey Test, $^*p < 0.05$, $^{**}p < 0.01$.

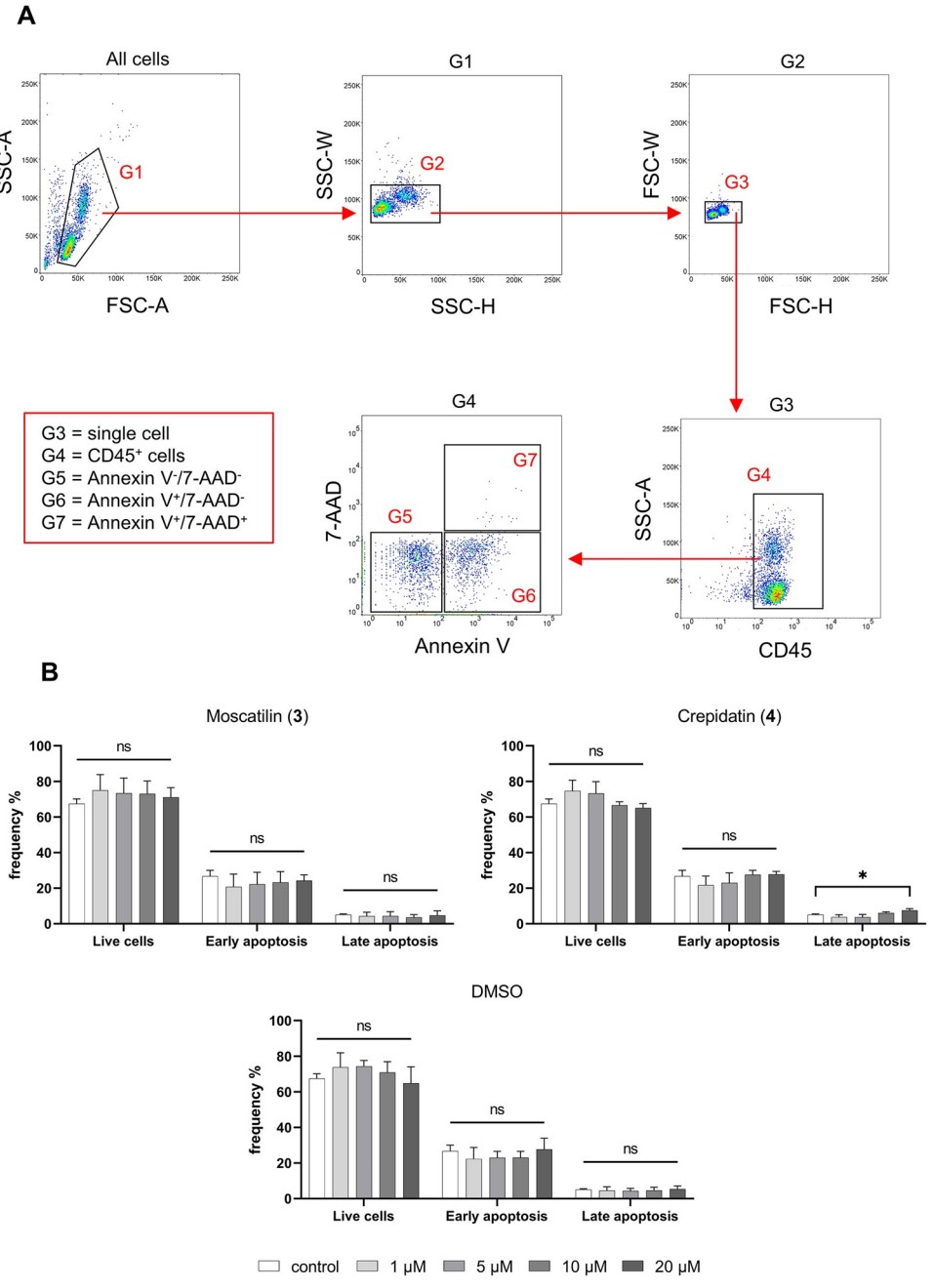

**Fig 5.** (A) Dot plots demonstrate gating strategy from flow cytometry in cytotoxicity staining with Annexin V and 7-AAD in human PBMCs used to obtain CD45 cells (G4) and determine the apoptosis state including live cells (G5), early (G6) and late apoptosis (G7). (B) Bar graphs show the mean frequency (%) changes of live cells and apoptosis state in human PBMCs treated with bibenzyl compounds **3**, **4** and DMSO, compared with only cells with medium. One-way ANOVA, corrected for multiple comparisons by Tukey Test, $^*p < 0.05$.

(Table 1), including ten phospho-molecules that were reported to be involved in LPS (TLR4)-mediated inflammatory responses (i.e., pNFkBp65, pSTAT1, pSTAT3, pSTAT5, CREB, pS6, p53, pH2AX, pAKT and pERK) and markers characterizing major circulating immune cell subsets such as T and B cells, myeloid cells (i.e., monocytes and dendritic cells) and natural

killer (NK) cells. As previously described [34], CyTOF data were pre-processed (i.e., de-barcoding, compensation and quality control) (Fig 6A). Subsequently, clustering analysis revealed a total of twenty clusters (Fig 6B and 6C). CD4$^+$ T cell cluster (cluster 9) showed the highest abundance among all immune cell type (Fig 6C, lower panel). In two of the three donors, increased abundance of CD56$^+$CD16$^+$Tbet$^+$ effector NK cells (cluster 4, Fig 6D), CD14$^-$CD16$^+$CD11c$^+$CXCR4$^+$ non-classical monocytes (cluster 7, Fig 6E) and CD14$^+$CD16$^{low}$ monocytes expressing co-stimulatory molecule CD86 (cluster 19, Fig 6F) was detected in LPS-treated PBMCs (LPS), compared to PBMCs without stimulation (no stimulation), whereas one of the three donors showed higher abundance of these clusters at the baseline (i.e., no stimulation condition) and a decreased frequency of these clusters after LPS stimulation. These results suggest high variation of immune cell profiling between healthy donors, and thus characterization in a much bigger cohort of healthy donors with new established antibody panel targeting a broader spectrum of immune cells is required. In addition, we also observed changes in immune profiling after treatment with both moscatilin (**3**) and crepidatin (**4**), compared to untreated and LPS-treated PBMCs (Fig 6D–6F). Similar to immune responses to LPS, these compound-induced phenotypic changes were different between donors, underlining high variation between primary blood cells from different healthy donors. Of note, in two of the three healthy donors, crepidatin (**4**) showed capacity to regulated LPS-induced phenotypic changes, determined by a reduction of the frequency of cluster 4, 7 and 19 compared to LPS-treated PBMCs (Fig 6D–6F). Again, no differences in phenotypes were detected in one of the three donors (donor 1), in both conditions with compounds and/or LPS. In contrast, we could not observe these phenotypic changes in LPS-treated samples after moscatilin (**3**) treatment. Next, we compared the expression of ten phospho-molecules involved in LPS (TLR4)-mediated inflammatory responses in all three differentially abundant clusters (cluster 4, 7 and 19), compared to other non-differentially abundant clusters (i.e., all other clusters). Interestingly, among all clusters non-classical monocytes (cluster 7) showed higher expression of pSTAT5 (Fig 6G and 6H), whereas higher expression of pERK was detected in CD86$^+$ monocytes (cluster 19) (Fig 6G and 6I). The expression of other phosphor-molecules was comparable between all clusters.

Taken together, our findings suggested that immunomodulatory effects of crepidatin (**4**) observed on LPS-stimulated monocytes (as shown in Fig 4) may be resulted from the reduction of CD14$^-$CD16$^+$pSTAT5$^+$ and/or CD14$^+$CD16$^{low}$CD86$^+$pERK$^+$ monocytes. These two monocyte sub-clusters seem to express TNF via different mechanism (i.e., phosphorylation of STAT5 or ERK). However, further study is required to confirm this hypothesis. In contrast, positive effects on monocytes provided by moscatilin (**3**) may be mediated by other monocyte-subsets and/or via different signaling pathway. For further investigation, new validated antibody panels for CyTOF analysis will be required. Nonetheless, our results also demonstrated high variation of immune responses between individuals, which most likely will occur in the real clinical application. Hence, prior to a clinical application of these active compounds or other natural products from *Dendrobium* plants, a precise monitoring of changes in phenotypes of multiple immune cell sub-populations, simultaneously, in a much bigger human cohort is required.

## Discussion

In this study we used LPS-induced PBMCs as a model for studying immunomodulatory activity of isolated known *Dendrobium* bibenzyl compounds. In line with our previous study [9], these findings confirm a common immunomodulatory effects of *Dendrobium* compounds. Their broad spectrum of activity on immune responses may have promising therapeutic

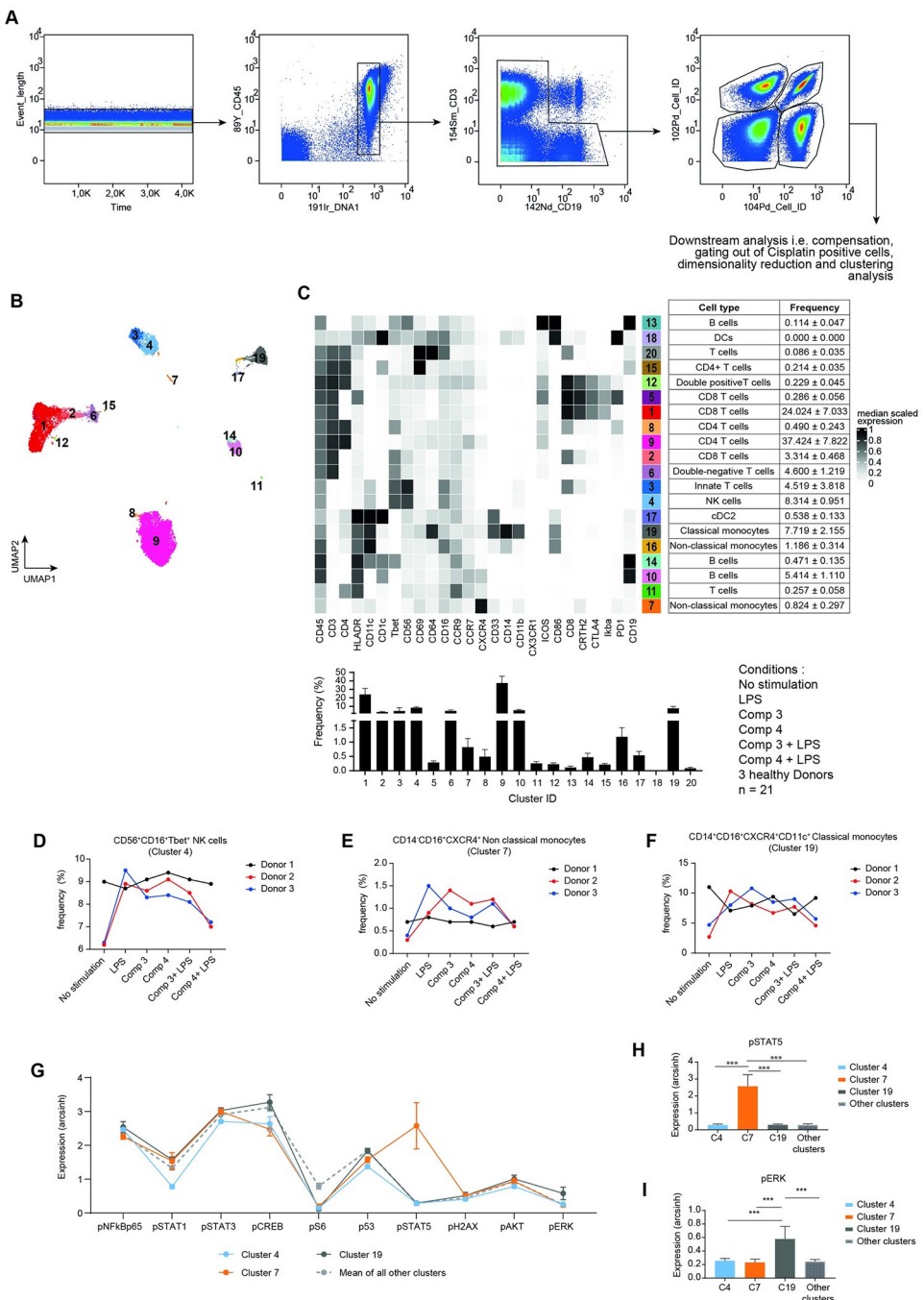

**Fig 6. Deep immune profiling using CyTOF.** (A) Gating strategy of the CyTOF data and downstream analysis such as selection of CD45[+] singlets cells, de-barcoding based on Boolean gating of palladium barcodes, selection of cisplatin[-] cells and clustering. (B) UMAP projection from all samples with 20 individually colored clusters representing diverse immune cell phenotypes, priorly defined by the FlowSOM algorithm (C) (top left) heatmap cluster illustrating the median expression levels of all markers analyzed with heat colors of expression levels scaled for each marker individually (to the 1st and 5th quintiles) (black: high expression, white: no expression); (top right) Cell type of each cluster and its respective frequency (mean ± SD); (lower panel) frequency plot (mean ± SD) of each cluster. (D-F) Frequency plots of differentially abundant clusters i.e., CD56[+]CD16[+]Tbet[+]CD45[+] NK cells (D), CD14[-]CD16[+]CXCR4[+] non-classical monocytes (E) and CD14[+]CD16[low]CD86[+] monocytes (F) between different PBMC-treated conditions from the three healthy donors. (G) Line graph of the arcsinh marker expression (mean ± SD) of the phosphor specific markers in cluster 4, 7 and 19, in comparison to the other clusters. (H) Bar graphs show higher expression level of pSTAT5 in cluster 7 compared to the other clusters. (I) Bar graphs show higher expression level of pERK in cluster 19

compared to the other clusters. Kruskal-Wallis test, corrected for multiple comparisons by Dunn's Test, ***$p < 0.001$, **$p < 0.01$, *$p < 0.05$.

potential in various diseases including inflammation. We showed the immunomodulatory effects of moscatilin (**3**) and crepidatin (**4**), indicated by the reduction of TNF-expressing monocytes in dose-dependent manner. Compound **3** and **4** showed immune modulatory effects which could be related to their structure-activity relationships (SAR). These compounds contain a core structure including one hydroxy group at C-4′ and three methoxy groups at C-3, C-5 and C-3′, which may relate to the inhibition of LPS-induced TNF expression in monocytes.

Deep-immune profiling using single-cell mass cytometry has further identified T-bet$^+$ NK cells (cluster 4), CD14$^-$CD16$^+$ non-classical monocytes (cluster 7) and CD86$^+$CD14$^+$ monocytes (cluster 19) as potential key populations that possibly associated with immunomodulatory effects of crepidatin. However, we failed to identify immune cell sub-populations that associated with the positive effects of moscatilin. We further identified possible mechanisms of action of crepidatin, which are the reduction of monocytes that highly expressed LPS-mediated phospho-molecules such as pSTAT5 (non-classical monocytes, cluster 7) and pERK (CD86$^+$ monocytes, cluster 19). Furthermore, the expression of co-stimulatory molecule CD86 on monocytes is required for activating lymphocytes [35, 36]. CD86 can bind two main receptors present on the surface of T lymphocytes, CD28 and cytotoxic T lymphocyte associated protein 4 (CTLA-4). Binding to CD28 results in T cell activation and can consequently enhance the immune response, whereas binding to CTLA-4 can lead to inhibition of T cell activation, thereby downregulating immunity [37, 38]. It remains unclear how crepidatin (**4**) can regulate T cell function via this CD86-expressing monocyte. In addition to reducing pSTAT5$^+$ and pERK$^+$ monocytes, we also detected a decreased abundance of Tbet$^+$ NK cells in LPS-stimulated cells treated with crepidatin. It has been shown that Tbet is an important transcription factor, which is essential for NK cell effector functions including sustained IFN-γ production as well as rapid production of perforin and granzymes for cytolytic activity [39].

## Conclusions

In summary, we have demonstrated herein dose-dependent immunomodulatory effects of bibenzyl compounds from *Dendrobium* species moscatilin (**3**) and crepidatin (**4**). For crepidatin, these positive effects are more likely associated with reduced abundance of monocyte sub-populations and NK cells, which potentially mediated the expression of TNF via pSTAT5 and pERK. Nevertheless, it remains to be investigated which cell populations associate with immunomodulatory effects of moscatilin and what is the possible mechanism of action of this compound. Our results show broad spectrum of activity on immune responses of Dendrobium compounds, which may lead to effective therapeutic potential of these compounds in complex disease conditions including inflammation. However, these results could also imply possible adverse effects in diverse immune cell types, and thus an evaluation of both therapeutic and adverse effects of such active compounds on multiple human immune cell populations using multi-parameter immune profiling method is required. Our findings also underline high variation of immune responses between individuals. Therefore, precise immune phenotypic characterization in a larger cohort using multi-parameter single cell analysis such as mass cytometry is recommended.

## Supporting information

**S1 Dataset.**
(XLSX)

## Acknowledgments

We would also like to acknowledge the assistance of the BIH Cytometry Core Facility (Charité–Universitätsmedizin Berlin, Germany).

## Author Contributions

**Conceptualization:** Chotima Böttcher, Boonchoo Sritularak.

**Data curation:** Virunh Kongkatitham, Adeline Dehlinger, Chatchai Chaotham, Kittisak Likhitwitayawuid, Chotima Böttcher.

**Investigation:** Virunh Kongkatitham, Adeline Dehlinger, Chotima Böttcher, Boonchoo Sritularak.

**Methodology:** Virunh Kongkatitham, Adeline Dehlinger, Chotima Böttcher, Boonchoo Sritularak.

**Visualization:** Virunh Kongkatitham, Adeline Dehlinger, Chotima Böttcher, Boonchoo Sritularak.

**Writing – original draft:** Virunh Kongkatitham, Adeline Dehlinger, Chotima Böttcher, Boonchoo Sritularak.

**Writing – review & editing:** Virunh Kongkatitham, Adeline Dehlinger, Chotima Böttcher, Boonchoo Sritularak.

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
