## [Decision Letter · Decision Letter 0]

10 Aug 2023

PONE-D-23-13279Immunomodulatory effects of bibenzyls from Dendrobium species on diverse human immune cell types under inflammatory conditionsPLOS ONE

Dear Dr. Böttcher,

Thank you for submitting your manuscript to PLOS ONE. After careful consideration, we feel that it has merit but does not fully meet PLOS ONE’s publication criteria as it currently stands. Therefore, we invite you to submit a revised version of the manuscript that addresses the points raised during the review process.

We look forward to receiving your revised manuscript.

Kind regards,

Syed M. Faisal, Ph.D.

Academic Editor

PLOS ONE

Journal Requirements:

   "This research project is supported by Second Century Fund (C2F), Chulalongkorn University to V.K. and B.S. and funded by Thailand Science research and Innovation Fund Chulalongkorn University (CU_FRB65_hea (57)_066_33_10). V.K. is grateful to C2F for conducting research abroad scholarship, Chulalongkorn University for a Ph.D. research abroad. "

Additional Editor Comments:

We appreciate your manuscript and the valuable insights it presents. However, prior to final acceptance, we kindly request that you address some key revisions to enhance the manuscript's quality and validity. Specifically, we recommend focusing on improving graph clarity, correcting inaccuracies related to compound effects, practicing caution in interpreting results, and ensuring missing values are included. Additionally, consider making minor adjustments, such as refining the abstract's focus, offering clarifications on prior studies, providing full forms for abbreviations, incorporating a schematic for compound purification, optimizing graph scaling and font size, rectifying legend inconsistencies, and accurately setting graph gaps. These revisions will undoubtedly contribute to elevating the manuscript's overall quality, aligning it more closely with our standards, and ensuring its readiness for final acceptance.

Reviewers' comments:

Reviewer's Responses to Questions

**Comments to the Author**

1. Is the manuscript technically sound, and do the data support the conclusions?

Reviewer #1: Yes

Reviewer #2: Partly

2. Has the statistical analysis been performed appropriately and rigorously? 

Reviewer #1: Yes

Reviewer #2: No

3. Have the authors made all data underlying the findings in their manuscript fully available?

Reviewer #1: Yes

Reviewer #2: Yes

4. Is the manuscript presented in an intelligible fashion and written in standard English?

Reviewer #1: No

Reviewer #2: No

5. Review Comments to the Author

Reviewer #1: Comments to the authors:

The manuscript entitled “Immunomodulatory effects of bibenzyls from Dendrobium species on diverse human immune cell types under inflammatory conditions” is interesting as it explores the potential of natural compounds in immunomodulation under inflammatory conditions.

The authors have put in good effort in the study, but the manuscript is not well written and there are many grammatical mistakes. Also, the flow and explanation of findings are not up to the mark.

There are multiple queries that are needed to be addressed:

1. In the abstract “Our study demonstrated board immunomodulatory effects…” correct broad. Also change the sentence to: Our study demonstrated broad immunomodulatory effects of Dendrobium compounds on multiple immune cell types including CD14+ monocytes.

2. In the abstract “This broad spectrum of activity on immune responses of Dendrobium compounds may lead to effectively therapeutic potential of these compounds in complex disease conditions including inflammation.” Replace effectively with effective. Also, this sentence is an exaggeration of the findings and hence can simply be written as “The broad spectrum of activity on immune responses of Dendrobium compounds may have promising therapeutic potential in various diseases including inflammation”.

3. The above 2 points are just an example of the grammatical errors and mistakes cited only from the abstract. The authors are recommended to get the whole manuscript checked by a writing expert.

4. Lines 48-51 can be put in the discussion section.

5. There are many sentences in the introduction section which have grammatical mistakes and thus change the context and meaning.

6. Line 86-87 says that LPS treatment is decreasing the TNF expression while from the results LPS is increasing the expression and the treatment with bibenzyls are reducing the expression of TNF.

7. What’s the rationale for taking 4 different Dendrobium species for isolating bibenzyls?

8. There is no information about the blood donor's age, gender, and health conditions from which PBMCs were isolated.

9. From Figure 3, it can be observed that treatment with compound 2 decreases the TNF in CD14+ monocytes even at 1µM concentration and is dose-dependent as well till 10 µM condition. Why didn’t the authors consider this compound for further studies when less concentration is showing significant results?

10. In Figure Legend 3, the authors should clearly state the treatment conditions. The current statement is confusing and misleading.

11. What’s the rationale for taking CD45+ cells for the cytotoxicity experiment? Why not the CD14+ monocytes taken for the study when the previous experiment shows the effect on these cells?

12. For deep immune profiling experiments, authors should consider PBMCs from more healthy donors. It is observed that the basal level of active immune cells of one of the donors is high without any stimulation.

13. Authors should consider isolating PBMCs from individuals with inflammatory conditions and then treating them with these compounds to substantiate their findings with more results.

Reviewer #2: Kongkatitham et. al., focuses on the immunomodulatory effects of seven bibenzyl compounds derived from Dendrobium plants. To do so the author’s mainly focuses on the effect of these compounds on various human peripheral blood mononuclear cells (PBMCs). They used flow cytometry and single-cell mass cytometry (CyTOF) to evaluate the impact of these compound on the cell population of several PBMCs. Moreover, they checked the phosphorylation status of multiple phosphor-proteins within the selected PBMCs types. The results clearly shows that LPS stimulation increases TNF expression primarily in CD14+ monocytes. Treatment of these cells with bibenzyl compounds exhibits inhibition of TNF expression in LPS-stimulated PBMCs. Among the seven compounds tested, author chose two bibenzyls, moscatilin and crepidatin, for further studies. Overall, this study provides a good preliminary assessment of the broad immunomodulatory effects of Dendrobium compounds on various immune cell types but fall short in addressing few key issues.

Major Comments

1. In Figure 3 – A graph showing a comparison of TNF expression in monocytes upon compound treatment—will help in better understating the data. Currently, it is challenging to make accurate assessments.

Could authors clarify the control condition? Furthermore, authors should elaborate on what occurs within the cells or with TNF expression in unstimulated cells upon compound treatment?

2. As authors stated “only two compounds (i.e., moscatilin (3) and crepidatin (4)) exhibited inhibitory effects in a dose-dependent manner and decreased LPS-induced TNF expression significantly at the concentration of 5, 10 and 20 μM (Fig. 3)”— however, based on the data in Fig. 3, this statement is not entirely accurate. For instance, in the presence of Aloifol (7), a marked and significant decrease in TNF expression can be observed at 5 μM compared to the LPS positive control. This also holds true for compounds 6, 5, and 2.

3. “Decreased abundance of CD56+CD16+Tbet+ effector NK cells (cluster 4, Fig.5D), CD14-CD16+CD11c+CXCR4+ non-classical monocytes (cluster 7, Fig. 5E) and CD14+CD16 low monocytes expressing co-stimulatory molecule CD86 (cluster 19, Fig. 5F) have been detected in LPS treated PBMCs after the treatment with crepidatin (4), compared to LPS-treated PBMCs”— appears overly assertive based on the available data. When examining the data for cluster 7, it's notable that donor 2 demonstrates no significant difference.

Moreover, the blood from donor 3 does not adhere to the observed trend whatsoever. In the case of cluster 4 and 7, no noticeable difference is observed in their frequencies. Additionally, contrary to the authors' assertion, an increase in the frequency of cluster 19 is observed, rather than a decrease. Considering the data, I would recommend that the authors incorporate additional donor data, accompanied by corresponding p-values, to elucidate the significance.

4. In Figure 5G, expression value without compound (control) is missing.

Minor comments

1. The authors should avoid the mentioning of number of biological replicates, and the concentration of the compounds used in the study in the abstract. The abstract should maintain a general focus, emphasizing the study's background, key questions addressed, the methodology employed for validation, and the ultimate outcomes.

2. In line 62-64 authors stated “However, these studies were mostly performed in cell lines or animal models, and very few studies were performed using primary human cell culture”. Does it mean that the prior studies were performed using cell line from animal models? Do specify. Additionally, provide references to support your statement.

3. In line 67-68: The full form of “TNF and IL-6” should be provided.

4. A schematic diagram of how these seven compounds were purified from the Dendrobium plants would be nice. It will help in to easily follow the steps used during purification.

5. Figure 2 and 3, Increase the graph scale and axis title font size. It is hard to read.

6. In Figure 2, the histograms represent all cell populations from G-5 to G-9, but the legends state otherwise. Please correct this discrepancy.

7. In Figure 5C, the selected gap for the bar graph is incorrect. The chosen gap value results in the omission of the error bar for group 7. Kindly redraw this bar graph with the appropriate gap. In heat map, scale correlating color with frequency is missing.

6. PLOS authors have the option to publish the peer review history of their article (what does this mean?). If published, this will include your full peer review and any attached files.

Reviewer #1: No

Reviewer #2: No

---

## [Author Response · Author response to Decision Letter 0]

14 Sep 2023

Reviewer #1: Comments to the authors:

The manuscript entitled “Immunomodulatory effects of bibenzyls from Dendrobium species on diverse human immune cell types under inflammatory conditions” is interesting as it explores the potential of natural compounds in immunomodulation under inflammatory conditions.

The authors have put in good effort in the study, but the manuscript is not well written and there are many grammatical mistakes. Also, the flow and explanation of findings are not up to the mark.

There are multiple queries that are needed to be addressed:

1. In the abstract “Our study demonstrated board immunomodulatory effects…” correct broad. Also change the sentence to: Our study demonstrated broad immunomodulatory effects of Dendrobium compounds on multiple immune cell types including CD14+ monocytes.

Response: 

We thank the reviewer for this and others comments and corrections. As suggested, we have corrected and also revised the abstract, in order to be more concise and coherent.

2. In the abstract “This broad spectrum of activity on immune responses of Dendrobium compounds may lead to effectively therapeutic potential of these compounds in complex disease conditions including inflammation.” Replace effectively with effective. Also, this sentence is an exaggeration of the findings and hence can simply be written as “The broad spectrum of activity on immune responses of Dendrobium compounds may have promising therapeutic potential in various diseases including inflammation”.

Response: 

We appreciate the reviewer´s comment and suggestion. We have revised this sentence, and (as suggested in the point 4.) we have moved it to the Discussion section as suggested. 

Original sentence: 

“Our findings suggest a broad spectrum of activity on immune responses of Dendrobium compounds, which may lead to effectively therapeutic potential of these compounds in complex disease conditions including inflammation.”

Revised sentence:

Line 344-345: “Their broad spectrum of activity on immune responses may have promising therapeutic potential in various diseases including inflammation.” 

3. The above 2 points are just an example of the grammatical errors and mistakes cited only from the abstract. The authors are recommended to get the whole manuscript checked by a writing expert.

Response: 

As recommended, the manuscript was checked by a writing expert, and has been revised accordingly.

4. Lines 48-51 can be put in the discussion section.

Response: 

As suggested, we have moved this sentence into the Discussion/Conclusion section (Line 377-380).

5. There are many sentences in the introduction section which have grammatical mistakes and thus change the context and meaning.

Response: 

Followed the suggestion of a writing expert, we have thoroughly revised the introduction section.

6. Line 86-87 says that LPS treatment is decreasing the TNF expression while from the results LPS is increasing the expression and the treatment with bibenzyls are reducing the expression of TNF.

Response: 

To avoid any misunderstanding we have revised the sentence as followed. 

Original sentence: 

“We demonstrated herein decreasing inflammatory responses of LPS-treated CD14+ monocytes, demonstrated by the reduction of inflammatory cytokine TNF.”

Revised sentence:

Line 82-83: “….Our results demonstrated that all seven bibenzyls could reduce the expression of TNF in CD14+ monocytes, induced by LPS treatment.”

7. What’s the rationale for taking 4 different Dendrobium species for isolating bibenzyls?

Response: 

To our experience, different Dendrobium species provide different yield (the amount of bibenzyl compounds per plant weight) of each bibenzyl compound. Moreover, a bibenzyl compound from different Dendrobium species often shows different effectiveness. We have chosen these 4 different Dendrobium species for isolating seven bibenzyls used in this study, on the basis of the results obtained from our previous studies (e.g., Sarakulwattana et al. Nat. Prod. Res. 2020; Phechrmeekh et al. J. Asian Nat. Prod. Res. 2012; Mittraphab et al. Nat. Prod. Commun. 2016).

8. There is no information about the blood donor's age, gender, and health conditions from which PBMCs were isolated.

Response:

We apologize for the missing information. In the revised manuscript, we have added these information in the Materials and Methods section (Line 132-133).

9. From Figure 3, it can be observed that treatment with compound 2 decreases the TNF in CD14+ monocytes even at 1 µM concentration and is dose-dependent as well till 10 µM condition. Why didn’t the authors consider this compound for further studies when less concentration is showing significant results?

Response: 

We agree with the reviewer that the compound 2 seems to be a good candidate for further analysis too. However, we have also observed that the compound 2 at the concentration of 5, 10 and 20 µM seems to be instable in our in vitro condition, e.g., changes in the colour of the culture medium and small amount of precipitation observed in culture medium at 10 µM (data not shown). This may possibly due to multiple hydroxyl groups, especially ortho-hydroxyl specie of the compound 2, and this may result in an increase of TNF expression that was observed at concentration 20 µM. For further use of this compound, its stability in vitro must be first evaluated.

10. In Figure Legend 3, the authors should clearly state the treatment conditions. The current statement is confusing and misleading.

Response:

We apologize for the mistakes. The Figure Legend has been revised as suggested.

11. What’s the rationale for taking CD45+ cells for the cytotoxicity experiment? Why not the CD14+ monocytes taken for the study when the previous experiment shows the effect on these cells?

Response:

We thank the reviewer for raising this point. Since PBMCs are used for our study, it is possible that bibenzyl compounds may also have effects and/or toxicity on other cell populations such as NK, T or B cells, which may then in turn activate and/or suppress monocytes. Therefore, to evaluate overall cytotoxicity of compounds on PBMCs, we measured cytotoxicity of bibenzyl compounds on CD45+ cells, which included all cell types in PBMCs. 

12. For deep immune profiling experiments, authors should consider PBMCs from more healthy donors. It is observed that the basal level of active immune cells of one of the donors is high without any stimulation.

Response:

We absolutely agree with the reviewer that our results not only showed immunomodulatory effects of bibenzyl compounds but also demonstrated high variation between donors, and thus to provide robust conclusion, further study in a larger cohort of healthy donors will be required. 

Our results show effects of bibenzyls on diverse immune cell population. Not only CD14+ monocytes (i.e., cluster 19) but also other immune cell sub-populations (i.e., CD14- non-classical monocytes and Tbet+ NK cells) could be affected by LPS and could possibly be further regulated upon compound treatment. Also, in this study we could not identify cell populations involved in potential mechanism of action of moscalitin, suggesting that further establishment of CyTOF analysis such as new antibody panels that target different cell populations and/or signalling pathway is necessary. And, as mentioned by the reviewer, more samples are required to obtain more reliable conclusion. However, it is technically challenging to measure more than 20 samples per barcode or multiple barcode sets (batches) using CyTOF due to its high batch-to-batch variation. A batch adjustment procedure (such as “Schuyler et al. Minimizing batch effects in mass cytometry data. Front Immunol. 2019”) is required for a direct comparison of runs performed across multiple batches over time. Of note, currently we are establishing new antibody panels and new CyTOF workflow containing batch-adjustment procedure.

However, we are convinced that publishing our current findings will provide other scientists new information and encourage them to consider an assessment of multiple cell populations, which may be affected upon treatment of natural products. Also, our results are important to demonstrate the high variation of human donors, which are different from animal models or human cell line, which are more homogenous. We encourage the community (natural product screening) to evaluate in primary human cells and using multi-parameter technology that will provide broader knowledge on potential effects of active natural product on multiple immune cell type. 

13. Authors should consider isolating PBMCs from individuals with inflammatory conditions and then treating them with these compounds to substantiate their findings with more results.

Response:

We thank the reviewer for suggestion. In our ongoing experiment, we plan to also evaluate therapeutic potentials of our compound in neuroinflammatory diseases such as multiple sclerosis using our new CyTOF workflow. However, as mentioned above, since donor-variation is high and will be even higher in inflammatory conditions, on the basis of our current results a much larger cohort of more than 50 patients/donor will be required. Single cell mass cytometry is challenging due to batch-to-batch variation and thus new workflow, including batch-to-batch normalization will be required. 

Reviewer #2: Comments to the authors:

Kongkatitham et. al., focuses on the immunomodulatory effects of seven bibenzyl compounds derived from Dendrobium plants. To do so the author’s mainly focuses on the effect of these compounds on various human peripheral blood mononuclear cells (PBMCs). They used flow cytometry and single-cell mass cytometry (CyTOF) to evaluate the impact of these compound on the cell population of several PBMCs. Moreover, they checked the phosphorylation status of multiple phosphor-proteins within the selected PBMCs types. The results clearly shows that LPS stimulation increases TNF expression primarily in CD14+ monocytes. Treatment of these cells with bibenzyl compounds exhibits inhibition of TNF expression in LPS-stimulated PBMCs. Among the seven compounds tested, author chose two bibenzyls, moscatilin and crepidatin, for further studies. Overall, this study provides a good preliminary assessment of the broad immunomodulatory effects of Dendrobium compounds on various immune cell types but fall short in addressing few key issues.

Major Comments

1. In Figure 3 – A graph showing a comparison of TNF expression in monocytes upon compound treatment—will help in better understating the data. Currently, it is challenging to make accurate assessments.

Could authors clarify the control condition? Furthermore, authors should elaborate on what occurs within the cells or with TNF expression in unstimulated cells upon compound treatment?

Response:

We thank the reviewer for valuable comments and suggestions. We apologies for unclear description of the results. In the revised manuscript, we have clearly described the different conditions including “Control” in both main text and figure legend.

Actually, in Figure 3 (or Figure 4 in the revised manuscript) we aimed to evaluate dose-dependent effects of seven bibenzyl compounds, in order to choose the most promising compound(s) for further deep-profiling study (shown in Figure 6 of the revised manuscript). Since it is technically very challenging to determine TNF expression along with phosphor-molecules, we have decided to further quantify the effects of two bibenzyl compounds (moscatilin and crepidatin) on the expression of ten phosphor-molecules involved in LPS-induced TNF expression. The results are shown in Figure 6, in which we attempted to elaborate on what occurs within the cells and/or with the LPS induction of TNF expression in different experimental conditions including unstimulated cells treated with only compounds. 

2. As authors stated “only two compounds (i.e., moscatilin (3) and crepidatin (4)) exhibited inhibitory effects in a dose-dependent manner and decreased LPS-induced TNF expression significantly at the concentration of 5, 10 and 20 μM (Fig. 3)”— however, based on the data in Fig. 3, this statement is not entirely accurate. For instance, in the presence of Aloifol (7), a marked and significant decrease in TNF expression can be observed at 5 μM compared to the LPS positive control. This also holds true for compounds 6, 5, and 2.

Response:

We apologies for unclear description of the results. For better understanding we have revised the relevant text on the interpretation of the results (Line 244-254). Briefly, we emphasize that we have detected immunomodulatory effects of all compounds (Line 244-245): “We detected a significant reduction in the frequency of TNF-expressing CD14+ monocytes treated with all bibenzyl compounds, except batatasin III (1)”. 

However, for further deep-immune profiling, we have chosen the compounds that provided dose-dependent effects, which are compounds 2, 3 and 4 (Line 248-249): “Nevertheless, only three compounds (4,5,4´-trihydroxy-3,3´-dimethoxybibenzyl (2), moscatilin (3) and crepidatin (4)) exhibited inhibitory effects in a dose-dependent manner (Fig 4).” Although the other compounds (i.e., 5, 6 and 7) also provided positive effects but they showed less dose-dependent. For example, compound 6 seems to provide higher effects at the concentration of 1 µM than 5 µM, and the concentrations of 10 and 20 µM seems to have comparable effects than 1 µM. 

Furthermore, we also provide a justification, why we did not choose the compound 2 for further investigation (Line 250-253): “However, we observed a sign of instability of compound 2 under our experimental conditions, e.g., changes in the color of culture medium, possibly due to multiple hydroxyl groups, especially the ortho-hydroxy. Of note, the compound 2 showed also a slight increase of TNF-expressing monocytes at the concentration of 20 µM. To prove the stability of the compound 2, further evaluation will be required.” 

3. “Decreased abundance of CD56+CD16+Tbet+ effector NK cells (cluster 4, Fig.5D), CD14-CD16+CD11c+CXCR4+ non-classical monocytes (cluster 7, Fig. 5E) and CD14+CD16 low monocytes expressing co-stimulatory molecule CD86 (cluster 19, Fig. 5F) have been detected in LPS treated PBMCs after the treatment with crepidatin (4), compared to LPS-treated PBMCs”— appears overly assertive based on the available data. When examining the data for cluster 7, it's notable that donor 2 demonstrates no significant difference.

Moreover, the blood from donor 3 does not adhere to the observed trend whatsoever. In the case of cluster 4 and 7, no noticeable difference is observed in their frequencies. Additionally, contrary to the authors' assertion, an increase in the frequency of cluster 19 is observed, rather than a decrease. Considering the data, I would recommend that the authors incorporate additional donor data, accompanied by corresponding p-values, to elucidate the significance.

Response:

We appreciate the reviewer’s comment and have revised the text in this section to avoid overstating the findings. Moreover, for readability as well as to allow readers to better judge our finding, we provide now source data sets of figure 4 (screening of seven bibenzyl compounds), 5 (cytotoxicity) and 6D-G (CyTOF analysis) . These data show that upon compound 4 treatment, LPS-treated PBMCs of both donor 2 and 3 demonstrate decreased frequency of cluster 7, in comparison to LPS-treated condition. 

As mentioned above (comment #12 of the reviewer #1), our results showed not only immunomodulatory effects of bibenzyl compounds on diverse immune cell populations, but also demonstrated high variation between donors, and thus (as also mentioned by the reviewer) further study with larger cohort of healthy donors will be required to have a robust conclusion. 

However, it is technically challenging to measure more than 20 samples per barcode or multiple barcode sets (batches) using CyTOF due to its high batch-to-batch variation. To precisely characterize immune responses of a bigger cohort of donors, further optimization of the CyTOF analysis workflow including a batch adjustment procedure is required. Also, since we could not unravel cell population(s) or cell signalling, which are associated with immunomodulatory effects of compound 3, establishing of new antibody panels that target broader spectrum of immune cell populations and/or specific signalling pathways is necessary. 

Nevertheless, we believe that publishing our current findings will provide other scientists new information and encourage them to consider multiple cell population in such analyses of natural products. Also, our results are important to demonstrate the high variation of human donors, which are different from animal models or human cell line, which are more homogenous. With our findings, we encourage the community (natural product screening) to evaluate in primary human cells and using multi-parameter technology that will provide broader knowledge on potential effects of active natural product on multiple immune cell type. 

4. In Figure 5G, expression value without compound (control) is missing.

Response:

The Figure 6G (or 5G in the original manuscript) show a comparison of phosphor-molecule expression in each differentially abundant clusters (i.e., cluster 4, 7 and 19) compared to other 17 defined clusters. For this graph, all cells from all donors and all conditions were pooled, and means and SDs were calculated. This plot aims to show differences in marker expression of each differentially abundant cluster, compared to other clusters. Therefore, expression value without compound is already included in the graph of each cluster. 

Minor comments

1. The authors should avoid the mentioning of number of biological replicates, and the concentration of the compounds used in the study in the abstract. The abstract should maintain a general focus, emphasizing the study's background, key questions addressed, the methodology employed for validation, and the ultimate outcomes.

Response:

We thank the reviewer for this comment and have revised the abstract accordingly.

2. In line 62-64 authors stated “However, these studies were mostly performed in cell lines or animal models, and very few studies were performed using primary human cell culture”. Does it mean that the prior studies were performed using cell line from animal models? Do specify. Additionally, provide references to support your statement.

Response:

We meant human cell lines and animal models. We have revised the manuscript, as suggested (Line 57).

3. In line 67-68: The full form of “TNF and IL-6” should be provided.

Response: 

We have added the full names of TNF and IL-6 in Line 62.

4. A schematic diagram of how these seven compounds were purified from the Dendrobium plants would be nice. It will help in to easily follow the steps used during purification.

Response: 

We thank the reviewer for this suggestion. In the revised manuscript, we have added the steps of isolation and purification of seven bibenzyl compounds from Dendrobium plants as Figure 3.

5. Figure 2 and 3, Increase the graph scale and axis title font size. It is hard to read.

Response: 

We have revised Figure 2 (or Figure 1 in the revised manuscript) and 3 (or Figure 4 in the revised manuscript) for readability.

6. In Figure 2, the histograms represent all cell populations from G-5 to G-9, but the legends state otherwise. Please correct this discrepancy.

Response: 

We thank the reviewer for pointing this mistake. We have revised the figure legend accordingly.

7. In Figure 5C, the selected gap for the bar graph is incorrect. The chosen gap value results in the omission of the error bar for group 7. Kindly redraw this bar graph with the appropriate gap. In heat map, scale correlating color with frequency is missing.

Response:

We have revised the figure as suggested.

---

## [Editor Report · Decision Letter 1]

18 Sep 2023

Diverse modulatory effects of bibenzyl from Dendrobium species on human immune cell responses under inflammatory conditions

PONE-D-23-13279R1

Dear Dr. Böttcher,

We’re pleased to inform you that your manuscript has been judged scientifically suitable for publication and will be formally accepted for publication once it meets all outstanding technical requirements.

Kind regards,

Syed M. Faisal, Ph.D.

Academic Editor

PLOS ONE
---

## [Editor Report · Acceptance letter]

27 Sep 2023

PONE-D-23-13279R1 

Diverse modulatory effects of bibenzyls from *Dendrobium* species on human immune cell responses under inflammatory conditions 

Dear Dr. Böttcher:

I'm pleased to inform you that your manuscript has been deemed suitable for publication in PLOS ONE. Congratulations! Your manuscript is now with our production department. 

Kind regards, 

on behalf of

Dr. Syed M. Faisal 

Academic Editor

PLOS ONE